# Correcting for volunteer bias in GWAS increases SNP effect sizes and heritability estimates

Sjoerd van Alten [1,2] ✉, Benjamin W. Domingue[3], Jessica Faul[4], Titus Galama[1,2,5] & Andries T. Marees[1]

Selection bias in genome-wide association studies (GWASs) due to volunteer-based sampling (volunteer bias) is poorly understood. The UK Biobank (UKB), one of the largest and most widely used cohorts, is highly selected. Using inverse probability (IP) weights we estimate inverse probability weighted GWAS (WGWAS) to correct GWAS summary statistics in the UKB for volunteer bias. Our IP weights were estimated using UK Census data – the largest source of population-representative data – made representative of the UKB's sampling population. These weights have a substantial SNP-based heritability of 4.8% (s.e. 0.8%), suggesting they capture volunteer bias in GWAS. Across ten phenotypes, WGWAS yields larger SNP effect sizes, larger heritability estimates, and altered gene-set tissue expression, despite decreasing the effective sample size by 62% on average, compared to GWAS. The impact of volunteer bias on GWAS results varies by phenotype. Traits related to disease, health behaviors, and socioeconomic status are most affected. We recommend that GWAS consortia provide population weights for their data sets, or use population-representative samples.

Genome-wide association studies (GWASs) have been instrumental in identifying genetic variation associated with various human traits, contributing to our understanding of the genetic basis of complex traits[1,2]. However, as with other associations derived from non-representative data[3], GWAS results could be affected by selection bias, since individuals who volunteer to participate are different from the underlying cohort-specific sampling population[3–10]. This type of bias, known as volunteer bias, may affect the internal validity of GWAS results, as study participation in itself can serve as a collider[4,11] from genotype to phenotype. We study whether volunteer bias affects GWAS findings for various phenotypes in the UK Biobank (UKB), one of the largest and most used GWAS cohorts.

Evidence suggests genetic studies are affected by non-random selection. For example, sex shows significant autosomal heritability in data sets that require active participation (23andMe and the UKB), but not in data sets that require more passive enrolment[6]. As no known biological mechanism could cause autosomal allele frequencies to differ between the sexes, such observed autosomal heritability of sex can be attributed to sex-differential participation bias. Further, genes are associated with study engagement[5,7,12,13]. However, it is unclear whether, how, and to what extent non-representativeness of samples biases (1) GWAS associations, and (2) various downstream analyses based on such GWAS results as an input (e.g., SNP-based heritabilities or gene-set tissue expression).

Non-random sample selection may bias single nucleotide polymorphism (SNP) associations in various directions, as we outline in detail in Supplementary Note S1. One possible scenario is Phenotype-related selection, which leads to attenuation bias. This results in smaller estimated SNP effect sizes, potential false negatives, and smaller SNP heritabilities. Under another scenario, Phenotype-

[1]Vrije Universiteit Amsterdam, Amsterdam, Netherlands. [2]Tinbergen Institute, Amsterdam, Netherlands. [3]Stanford University, California, US. [4]University of Michigan, Ann Arbor, US. [5]University of Southern California, Dornsife Center for Economic and Social Research and Department of Economics, California, US. ✉e-mail: s.j.d.van.alten@vu.nl

genotype-related selection, collider bias occurs: a correlation between the SNP and the phenotype appears even if the SNP does not influence the phenotype. This scenario could result in false positives when the true SNP effect size is zero or can result in incorrect effect sizes (possibly even of the opposite sign) for SNPs that do have an effect on the phenotype.

The UKB, an essential data source for GWAS given its large sample size ($N \approx 500,000$) and deep phenotyping[14], suffers from selective participation: only 5.5% of UK citizens who received an invitation participated. Those that did are more likely to be older, female, and of higher socioeconomic status compared to the invited population[15]. Here, we use inverse probability (IP) weights to estimate genetic associations in the UKB that are robust to volunteer bias. We do this by conducting inverse probability weighted GWAS (WGWAS) for 10 phenotypes and assessing the effects of volunteer bias on GWAS results by comparing these WGWAS results with unweighted GWAS results estimated for the same phenotypes.

In an earlier study, we compared UKB and UK Census data to demonstrate how selective participation in the UKB results in substantial biases in various phenotype-phenotype associations; these biases can be quite severe and even lead to associations that have the incorrect sign[3]. We also constructed IP weights, designed to correct for volunteer bias in these associations. These IP weights were estimated using a subsample of the UK Census data, selected to be representative of all UK citizens who received an invitation to participate in the UKB (the UKB-eligible population). These IP weights are precisely estimated and capture an average of 87% of the volunteer bias in various estimated phenotype-phenotype associations[3]. Thus, by weighting the UKB, we can substantially remove bias in association estimates due to volunteering.

Our work builds on other attempts to correct for volunteer bias in the UKB[16–18], that used the Health Survey of England (HSE), a smaller and less representative survey data set. We instead used UK Census data to construct IP weights, which led to several improvements. First, the UK Census, with a response rate of 95%, is highly representative of the UK population and has a much larger sample size than any data source previously used. By contrast, only 68% participated in the HSE, such that representativeness of the HSE itself is not guaranteed. Second, using detailed UK Census geographic location information, we precisely restricted the UK Census data to the UKB-eligible population: those aged 40–69 between 2006 and 2010 who lived within the sampling radius of any of the 22 UKB assessment centres. Prior studies have not done that. Third, we estimated the IP weights using predictors of selection bias that were missing in previous analyses, most importantly, region of residence, which is one of the strongest predictors of selection into the UKB[3]. Fourth, our IP weights are developed using a sample of 687,489 versus 22,646 individuals by Schoeler et al.[18], who also explored the consequences of volunteering on genetic analyses. Last, our IP weights are available for the full UKB, rather than a subsample, increasing power. This allows us to construct the largest WGWAS in respondents of European ancestry to date ($N = 376, 900$ versus $N \sim 283,000$ in Schoeler et al.[18]). Combined, these factors helped us develop IP weights that are both more meaningful (better matching the UKB-eligible population) and more precisely estimated (larger sample sizes / more relevant predictors).

As a consequence, we believe our weights are currently the best weights available to capture volunteer bias in the UKB. In support of this, we demonstrate that the UK Census-based weights have an SNP-based heritability of 4.8% (s.e. 0.8%), whereas weights used in previous research based on these smaller data sets had substantially smaller heritability (0.9%, s.e. 0.5%)[18]. Our main results compare WGWAS to GWAS and suggest that volunteer bias is of even greater importance to GWAS than has previously been shown[18]. Further, in our analyses, SNP effect sizes and SNP-based heritabilities increase after taking volunteer

bias into account, whereas in earlier work, effect sizes mostly decreased[18].

## Results

IP weights are available for ~98% of all 502,500 UKB respondents (see Methods). After various quality control (QC) steps and restricting to respondents of white European ancestry, our sample consists of 376,900 respondents (see Methods and Supplementary Fig. S1). This sample closely resembles the UKB sample typically used in GWAS analyses. We selected 10 phenotypes related to health and social science outcomes, all collected at baseline. Age at first birth (AFB) and breast cancer were studied in females only. Supplementary Data 1 summarises these phenotypes before and after IP weighting. Weighting changes their means and standard deviations. For example, the UKB oversampled those with more education: UKB respondents received an average of 13.8 years of education (s.e.= 0.007; SD = 4.91), whereas the mean weighted average is 13.0 years (s.e. = 0.011; SD = 5.0). The sample size for all ten phenotypes is larger than 140,081, with an average N of 320,235 and maximal N of 376,900. Supplementary Note S2 outlines our coding procedures for each phenotype.

### IP weights capture the genetic component of healthy volunteer bias in the UKB

To assess whether our IP weights capture volunteer-based selection that may affect phenotype-genotype associations, we first consider GWAS with the IP weights as the phenotype (See "Methods"). We are interested in the unconditional association between the genome and the IP weights, since controlling, at this stage, for a set of first principal components (PCs) of the genetic data, may inadvertently remove genetic signal: PCs capture the largest sources of genetic variation in the data and may therefore be associated with selection (volunteering) into the UKB. For example, the probability to participate in the UKB differs by region[3], and is therefore likely to correlate with population stratification (and thus the first PCs of the genetic data). Controlling for population structure in the GWAS on the IP weights may, therefore, remove genetic signals that is relevant to understand the type of bias that the IP weights correct for. In this unconditional GWAS, we found 7 independent genome-wide significant loci (Supplementary Fig. S2) and a SNP-based heritability of 3.61% (s.e. 0.26%, LD-score intercept: 1.309 [0.009]).

Indeed, an LD-score intercept larger than one suggests our weights are associated with population stratification. To test whether the IP weights capture more than just population stratification, we reran the GWAS on IP weights with a standard set of controls, including the first 20 principal components (see "Methods"). The resulting heritability of the weights decreased but remained substantial at 2.28% (s.e. 0.17 %). The intercept was now substantially closer to one (1.038 [0.007]). To verify that any remaining heritability was not inflated due to residual population stratification or cryptic relatedness, we accounted for these by re-estimating this GWAS using a linear mixed model (BOLT-LMM)[19], ($h^2 = 2.24\%$, s.e. = 0.16%). Accounting for residual population stratification and cryptic relatedness did not alter our results. The genetic correlation of the GWAS results estimated with the linear model, using PLINK, and the linear mixed model, using BOLT-LMM, was indistinguishable from 1 ($r_G = 1.0009$, $s.e. = 0.0018$).

The heritability, as estimated using WGWAS with controls, was 4.8% (s.e. 0.8%, ld-score intercept 1.009 [0.006]). This weighted heritability is directly comparable to the one reported in a previous study that weighted GWAS associations, based on weights derived from the HSE[18]. The heritability of these HSE-based weights was much lower: 0.9% (s.e. 0.5%; LD-score intercept 1.055).

The substantially higher heritability of our UK Census-based IP weights, compared to previous efforts, confirms our prior that the UK Census is a better data source to estimate UKB IP weights on, for

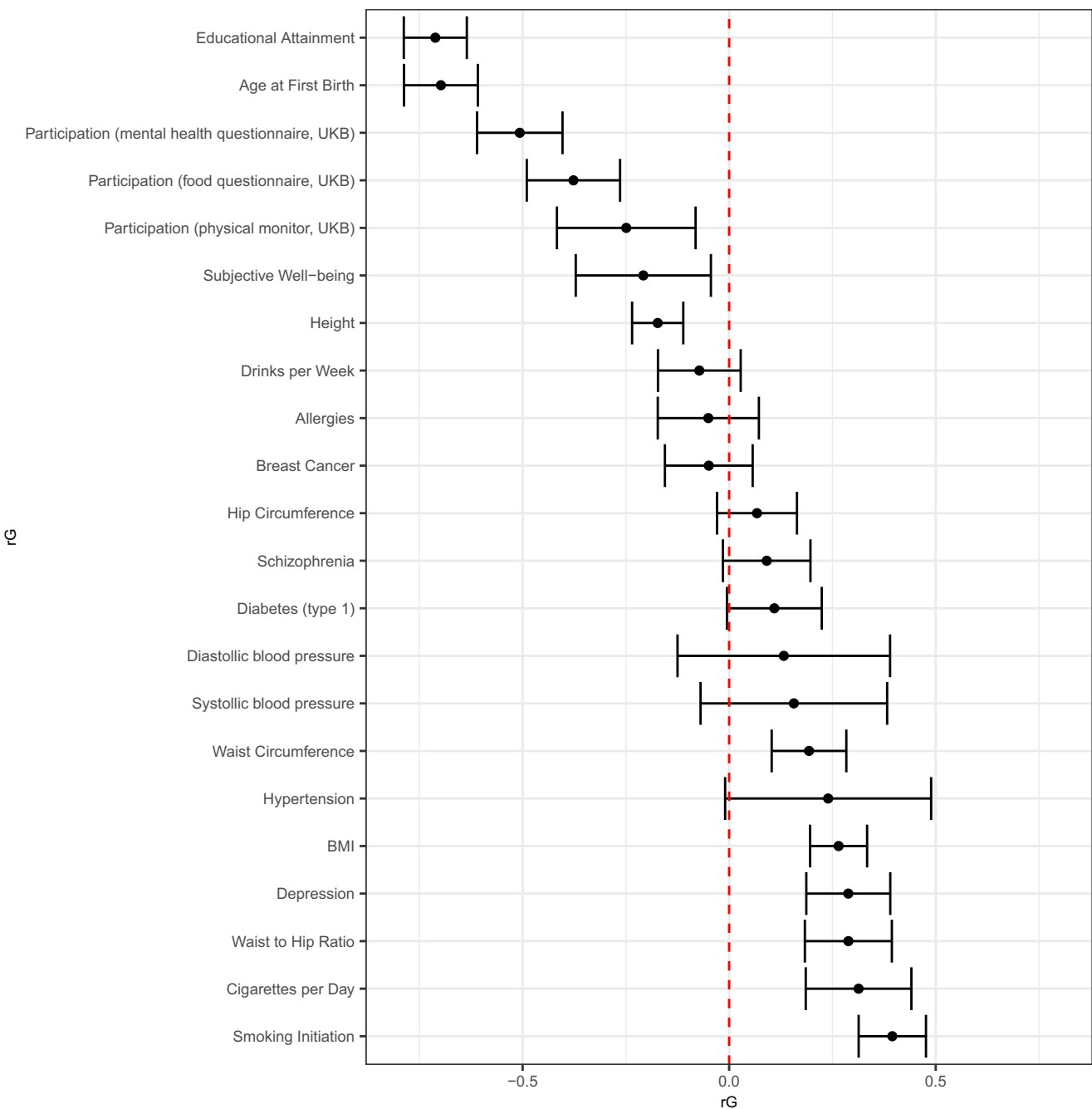

**Fig. 1 | Genetic correlations between IP weights and various phenotypes.** Correlations are based on our (unconditional) GWAS of the IP weights and on existing GWAS results for various phenotypes (see Supplementary Data 3 for the sources and sample sizes of each respective GWAS). The sample size for the GWAS on the IP weights was $N$=376,900. Each point shows the estimate of the genetic correlation between the weights and the respective phenotype, as estimated by LD-score regression. Bonferroni-corrected 95% confidence intervals for the 22 hypotheses tested are shown around each estimate. Respondents with a lower probability of participating in the UKB are assigned a higher IP weight. Thus, a negative (positive) genetic correlation between the GWAS on the IP weights and a phenotype implies that individuals with a higher genetic propensity for that phenotype also have a higher (lower) genetic propensity to volunteer for the UKB.

several reasons such as (1) larger sample size, (2) more relevant variables included in weights estimation such as region, (3) larger representativeness due to a response rate of > 95%, and (4) the ability to precisely restrict the target data to the population invited to participate in the UKB[3].

We next interpret associations of our GWAS on the IP weights (without controls) in more detail. The qq-plot for the IP Weights GWAS shows an early lift-off ($\lambda$ = 1.55; Supplementary Fig. S3), suggesting that the IP weights are highly polygenic and that volunteer bias impacts genetic associations across the genome. Figure 1 shows

strong and statistically significant genetic correlations between the IP weights and various phenotypes (see "Methods"). The observed pattern is consistent with the IP weights capturing healthy volunteer bias, as they reflect that those in better health and of higher socioeconomic status (e.g., higher years of education) are more likely to participate in the UKB. For example, SNPs associated with a higher IP weight − i.e., with individuals that are underrepresented within the UKB − are also associated with lower education ($rG$ = − 0.711 [0.025]), higher BMI ($rG$ = 0.265 [0.023]), and a higher likelihood of mental disorders (e.g., Depression $rG$ = 0.288 [0.033]). Overall, these

**Table 1 | Comparison of weighted and unweighted GWAS results (top hits [$p < 10^{-5}$] only)**

| Phenotype | Coefficient [95% CI] | P | N |
|---|---|---|---|
| Years of Education | 1.109 [1.087;1.131] | $5.16 \times 10^{-21}$ | 504 |
| BMI | 1.091 [1.068;1.115] | $2.89 \times 10^{-13}$ | 259 |
| Severe Obesity | 1.082 [1.028;1.137] | $3.00 \times 10^{-3}$ | 259 |
| Height | 1.021 [1.014;1.028] | $3.83 \times 10^{-9}$ | 1967 |
| Drinks Per Week | 1.183 [1.054;1.312] | $7.05 \times 10^{-3}$ | 30 |
| Breast cancer | 0.839 [0.800;0.878] | $4.00 \times 10^{-15}$ | 510 |

Each row shows the coefficient (and 95% confidence interval) for a bivariate regression with the weighted SNP effect as the dependent variable and the unweighted SNP effect as the independent variable. A coefficient larger than one implies that WGWAS increases GWAS effect sizes on average (i.e., volunteer bias leads to an underestimate of the association in GWAS). A coefficient smaller than one implies that WGWAS shrinks effect sizes on average. P-values are for the null hypothesis that this coefficient equals one, based on a two-sided t test. The last column shows the number of SNPs that are included in the regressions: only independent lead SNPs from GWAS studies that did not include the UKB are included (see Methods for additional detail).

findings suggest that volunteer bias, as captured by the IP weights, influences the genetic profile contained in the UKB.

Supplementary Note S3 provides various follow-up analyses on the IP weights GWAS. Any genome-wide significant loci in extant GWAS analyses that include the UKB and that were also significant in our IP weights GWAS should be considered suspect. To aid researchers, we list all 408 suggestive top hits from our IP weights GWAS ($P < 5 \cdot 10^{-5}$, based on a two-sided Z-test) in Supplementary Data 2.

## WGWAS estimation corrects genetic associations for volunteer bias

We first investigate the relation between WGWAS and GWAS SNP effects for previously identified top hits for each phenotype. We define a top hit as having $p < 10^{-5}$ in a publicly available well-powered GWAS ($N > 200,000$, see Supplementary Data 4) that did not include the UKB (see "Methods"). Because well-powered GWAS that do not include UKB data are not available for every phenotype, we could only perform these analyses for 6 out of the 10 phenotypes. Table 1 shows the coefficient of a regression of the effect sizes of these top SNPs estimated through WGWAS on the effect sizes of the same SNPs estimated through GWAS. These coefficients are significantly larger than one, except for breast cancer. Thus, for most phenotypes, correcting GWAS for volunteer bias through WGWAS results in more predictive effect sizes, i.e., effect sizes that lie further from the null, which is consistent with selection bias, here taking the form of attenuation bias. Such attenuation is to be expected when selection into the data is based on the phenotype, rather than the genotype (see Supplementary Note S1).

Education, BMI, severe obesity, and drinks per week are most affected by this type of phenotype-related volunteer bias: correcting for volunteer bias results in an increase of the SNP effect sizes by 10.9% for years of education, 9.1% for BMI, 8.2% for severe obesity, and 18.3% for drinks per week. By contrast, estimating a WGWAS of height also results in larger effects, but the overall effect is small: a 2.1% increase in the effect sizes. This is consistent with evidence that height plays a relatively small role in whether individuals volunteer to participate (see Fig. 1 and Supplementary Data 1).

Breast Cancer is the only phenotype for which we find a significant shrinkage of SNP effect sizes (Table 1), with a coefficient on the regression of 0.839. Hence, not taking volunteer bias into account inflates genetic effect sizes for previously identified top hits for breast cancer, which implies that some of these previously identified SNPs may have overestimated effect sizes. As breast cancer is a binary phenotype that is oversampled in the UKB, such an overestimation is expected under phenotype-related selection (see Supplementary Note S1). While oversampling of a disease-related phenotype is at odds

with the idea of healthy volunteer bias, it could result from older women being more likely to volunteer, in combination with the increasing prevalence of breast cancer with age[20].

Table 2 provides additional comparisons of WGWAS and GWAS results for all ten phenotypes using WGWAS and GWAS effect sizes for all SNPs, i.e., not restricted to previously identified top hits ($p < 10^{-5}$). The first column shows the genetic correlation between the unweighted and weighted GWAS effect sizes (see "Methods"). The correlation is positive in all cases and close to one for most phenotypes, but differs statistically significantly from one (at a Bonferroni-corrected level of $p < 0.05$) for 6 out of 10 phenotypes (indicated by *). The lowest congruence between weighted and unweighted SNP associations is found for T1D ($rG = 0.66$) and Breast Cancer ($rG = 0.80$).

We use the standard errors of WGWAS (GWAS) to estimate the effective sample size in columns 2 and 3 of the table (see "Methods"). Based on this comparison, the effective sample size in WGWAS shrinks by 62% compared to the effective sample size in GWAS when averaged across all phenotypes. To put this shrinkage into perspective: a hypothetical UKB of size 143,222 that is representative would have equivalent power when estimating GWAS as compared to the actual UKB sample of 376,900 respondents, illustrating the loss in power that is a consequence of relying on volunteers. Related, column 4 shows an increase in the standard errors for each phenotype after applying WGWAS, which ranges from 40.0% for breast cancer to 87.0% for T1D. The need to correct for volunteer bias in the UKB increases standard errors (since a weighted estimator is typically less efficient than an unweighted one), and hence decreases the effective sample size. Intuitively, volunteering leads to an oversampling of similar-type individuals, reducing the effective sample size of a representative sample through IP weighting. Thus, when correcting genetic associations for selection bias using IP weighting, researchers face a form of bias-variance tradeoff: unweighted estimates seem more precise but are potentially biased, whereas weighted estimates will be less precise, but are less biased.

Columns 5 and 6 of Table 2 document a decrease in genome-wide significant SNPs from WGWAS relative to GWAS. Here, we only consider independent loci as identified through clumping of WGWAS (GWAS) summary statistics (see Methods). For example, the number of genome-wide significant lead SNPs in our BMI GWAS is 1205, whereas it is 127 in the corresponding WGWAS. These newly insignificant SNPs may indicate false positives in the current GWAS literature, but may also be a result of the increased standard errors that are a feature of WGWAS.

Column 7 shows that WGWAS can find signals previously deemed insignificant in GWAS: this column shows hits that are unique to WGWAS, i.e., SNPs not genome-wide significant in GWAS but genome-wide significant in WGWAS. For 6 out of the 10 phenotypes we tested, correcting for volunteer bias results in such unique hits. However, not all these SNPs should be considered discoveries. For example, an SNP could be just shy of significance in GWAS and then cross the threshold of genome-wide significance in WGWAS, not necessarily due to a significant volunteer bias correction, but simply due to chance. To address this, we use a Hausman test to calculate p-values for the null hypothesis that the effect sizes in weighted and unweighted GWAS are the same (termed $P_H$, see Methods). For each phenotype, the qq-plots of $P_H$ are shown in Supplementary Fig. S4. As described in the next section, using this very strict method of testing, we find a total of four new loci that we consider newly discovered by WGWAS (column 8 in Table 2).

## Correcting for volunteer bias results in the discovery of new loci that were previously attenuated in GWAS

We consider an SNP a newly discovered locus if it is insignificant in GWAS, significant in WGWAS (at $P < 5 \times 10^{-8}$) and if there is sufficient evidence that WGWAS estimates a different effect size for this SNP,

**Table 2 | Comparison of weighted and unweighted GWAS results**

| Phenotype | (1)<br>$r$ | (2)<br>$N_{eff}^{GWAS}$ | (3)<br>$N_{eff}^{WGWAS}$ | (4)<br>Increase S.E.s | (5)<br>sig. hits GWAS | (6)<br>sig. hits WGWAS | (7)<br>unique<br>hits WGWAS | (8)<br>New loci |
|---|---|---|---|---|---|---|---|---|
| Age at First Birth | 0.976 (0.013) | 139,093 | 51,949 | 71% | 30 | 3 | 2 | 0 |
| BMI | 0.992 (0.005) | 372,969 | 135,238 | 77% | 1205 | 127 | 5 | 0 |
| Breast cancer | 0.803*(0.038) | 197,857 | 90,492 | 40% | 45 | 8 | 4 | 1 |
| Drinks per Week | 0.936*(0.019) | 265,696 | 96,008 | 83% | 23 | 4 | 0 | 0 |
| Self-rated health | 0.973*(0.009) | 372,714 | 136,982 | 82% | 101 | 6 | 0 | 0 |
| Height | 0.993 (0.003) | 374,175 | 151,328 | 61% | 5114 | 1453 | 22 | 0 |
| Physical activity | 0.866*(0.031) | 334,570 | 123,017 | 75% | 3 | 0 | 0 | 0 |
| Severe Obesity | 0.949*(0.018) | 373,834 | 136,396 | 75% | 23 | 1 | 0 | 0 |
| Type 1 Diabetes | 0.660*(0.057) | 373,786 | 132,605 | 87% | 69 | 37 | 15 | 3 |
| Years of Education | 0.988 (0.006) | 392,433 | 160,707 | 64% | 331 | 49 | 3 | 0 |

Comparisons use all UKB SNPs in HapMap3 (1,025,058 in total). The first column shows the genetic correlation $r$ between GWAS and WGWAS results, estimated through LD-score regression (see "Methods"). The second and third columns show the effective sample sizes (see Methods) for GWAS and WGWAS. WGWAS increases standard errors by the percentage shown in column 4 (Increase S.E.s). column 5 shows the number of genome-wide significant SNPs for each trait in GWAS ($P < 5 \cdot 10^{-8}$, based on a two-sided $t$ test); column 6 shows this in WGWAS; column 7 indicates how many of these genome-wide significant SNPs in WGWAS are unique. i.e., these SNPs have $P < 5 \cdot 10^{-8}$ in WGWAS, but $P \geq 5 \cdot 10^{-8}$ in GWAS. Last, column 8 shows how many of these WGWAS-tagged new loci are unique hits, as indicated by a Hausman test that tests for genome-wide significance in the difference in the effect size as estimated through GWAS and WGWAS, a stringent test (see Methods). Thus, these loci were insignificant in GWAS, significant in WGWAS, and the difference in the effect sizes was genome-wide significant ($P_H < 5 \cdot 10^{-8}$). * indicates values significantly different from one at a Bonferroni-corrected level of 5% significance, correcting for multiple hypothesis testing across ten phenotypes, i.e., ($p < 0.05/10 = 0.005$), based on a one-sided $Z$-test.

compared to the one estimated by GWAS by using a Hausman test that tests for genome-wide significance ($P_H < 5 \times 10^{-8}$) in the difference of the effect sizes. Although very stringent, we identify a total of four independent loci that satisfy all these criteria: three for T1D, and one for breast cancer (Supplementary Data 5 and 6). For example, lead SNP rs17186868 is insignificant for T1D in GWAS ($\hat{\beta} = -0.0012$, $s.\hat{e}. = 0.00080$, $P = 0.13$), but is genome-wide significant in WGWAS ($\hat{\beta} = -0.0052$, $s.\hat{e}. = 0.00082$, $P = 2.64 \cdot 10^{-10}$). Further, the difference in these point estimates is highly significant ($P_H = 1.28 \cdot 10^{-91}$). The other two newly identified genome-wide significant lead SNPs are rs341988 and rs12522568.

Hence, for T1D, volunteer bias results in missing several genome-wide significant loci. A comparison of the Manhattan plots for GWAS and WGWAS for T1D visually demonstrates that weighting alters which loci become significant and which ones become insignificant for T1D (Fig. 2). For breast cancer, WGWAS similarly results in the discovery of one new locus, with lead SNP rs2306412.

We further explored these four newly identified lead SNPs for T1D and breast cancer in the GWAS catalogue and in previous publicly available GWAS results for these phenotypes, which were conducted in data sets that did not include the UKB (Supplementary Note S4). These four loci have not previously been identified as being associated with these phenotypes, and are thus potentially novel. However, rs17186868, which we associated with T1D in WGWAS but not in GWAS, has previously been found to be associated with BMI-adjusted waist circumference. Given that they have not been previously found in the literature, one concern may be that these associations are due to specific modelling assumptions. In Supplementary Note S5, we show that these findings are robust to various modelling assumptions in the UKB: the difference between weighted and unweighted effect sizes replicates when (1) removing all control variables from the model, (2) moving from a linear probability model to a logit model, and (3) changing the specification that underlies the estimation of our IP weights.

We flag these novel loci with caution, given the lack of evidence of signal of these SNPs in previous work. We would not encourage, for example, direct biological follow-up related to these loci but rather note them in case they become of interest in future GWAS work, or when future GWAS replicates these loci for these traits. Further, these findings again illustrate that weighting GWAS results for volunteer bias may lead to substantially different results.

## SNP heritability estimates become larger after correcting for volunteer bias

Results presented in subsection 2.1 suggest that the genetic influences of volunteer bias are highly polygenic. This suggests that volunteer bias can affect SNP associations throughout the genome in subtle ways that cannot be detected individually (due to a lack of power), but that can substantially impact downstream analyses of GWAS results that aggregate SNP effects across the genome. In the remainder, we investigate how weighting GWAS results affects various downstream findings.

We estimated SNP-based heritabilities − the proportion of phenotypic variance explained by SNPs − using LD-score regression (see "Methods") based on GWAS/WGWAS. We use the effective sample sizes (see Table 2) to account for the increased estimation error of WGWAS vis à vis GWAS[21]. Results are summarised in Table 3.

For most phenotypes, correcting for volunteer bias by WGWAS results in substantial increases in SNP-based heritability estimates, consistent with the increase in effect sizes after weighting (Table 1). As in section 2.2, weighting matters most for T1D and breast cancer. For T1D, the SNP-based heritability increases from 0.54% in GWAS to 4.32% in WGWAS, a large and highly statistically significant increase ($P = 1.63 \cdot 10^{-41}$). For breast cancer, the heritability almost doubles from 2.59% to 5.12% ($P = 2.37 \cdot 10^{-8}$). Most other phenotypes also have higher heritabilities. For example, education has a heritability of 14.8% in GWAS, and this increases to 17.8% when volunteer bias is taken into account ($P = 2.07 \cdot 10^{-9}$). Drinks per week, severe obesity, age at first birth, and self-rated health also show substantial increases in estimated SNP heritabilities. This is consistent with phenotype-related selection (Supplementary Note S1). By contrast, Height, BMI, and Physical Activity do not show significant changes in heritability.

In LD-score regression, an intercept greater than 1 may be indicative of bias due to population stratification or cryptic relatedness[22]. For our unweighted GWASs, we find intercepts larger than 1 for years of education, BMI, height, self-rated health, and AFB, as is common for these phenotypes[23–25] (see Table 3, column 5). After weighting, the intercept moves closer to one; for self-rated health and AFB, it is statistically indistinguishable from one (see Table 3, column 6). Hence, WGWAS may have the additional advantage of reducing bias arising from population stratification.

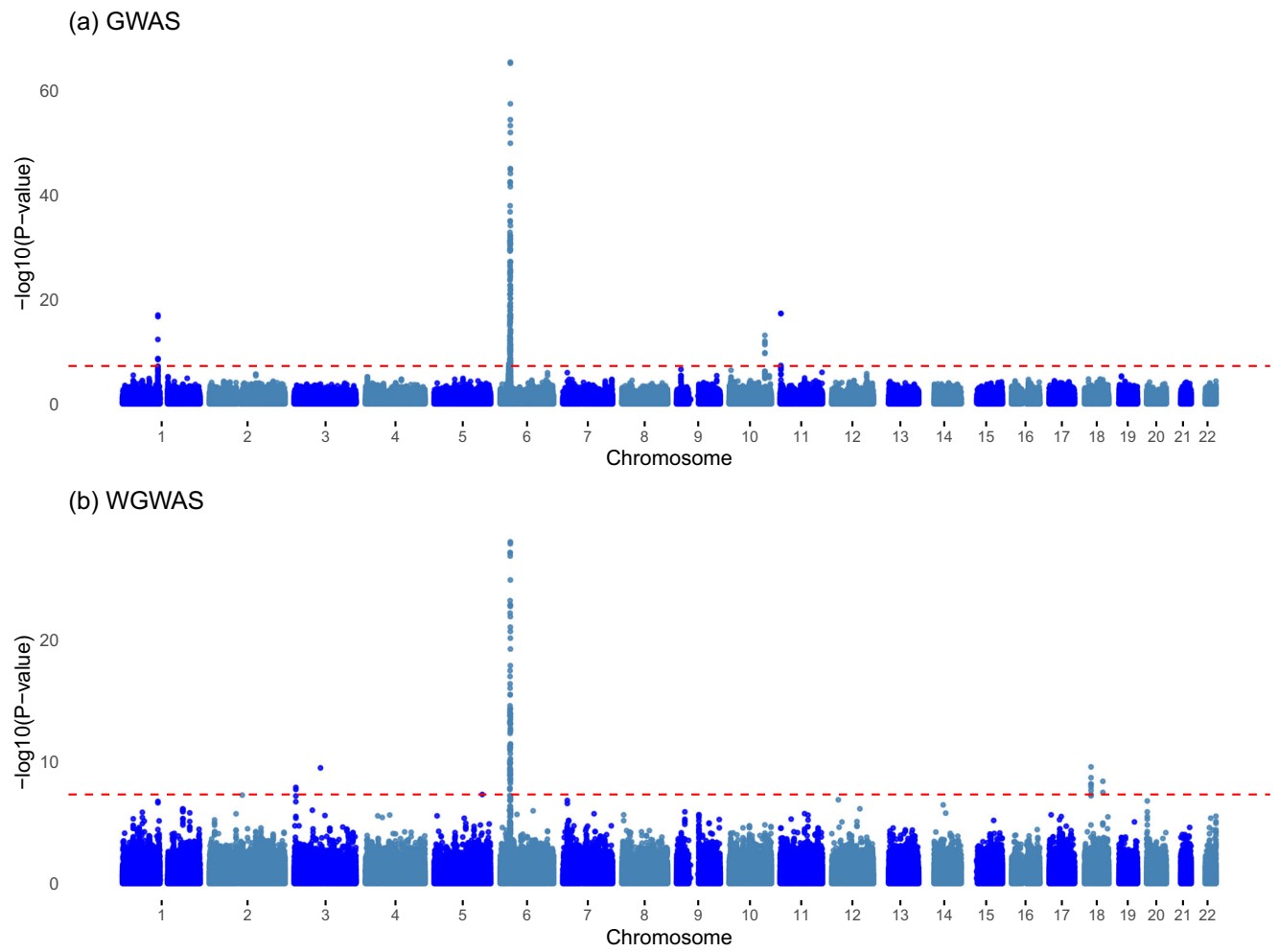

**Fig. 2 | Manhattan plot of GWAS and WGWAS results for type 1 diabetes.** Panel (**a**) displays the results as estimated by GWAS. Panel (**b**) displays the results as estimated by WGWAS. *P*-values are based on two-sided *Z*-tests.

## Table 3 | SNP-based heritabilities for GWAS and WGWAS

| Phenotype | GWAS $h^2$ (SE) | WGWAS $h^2$ (SE) | P | GWAS Intercept (SE) | WGWAS Intercept (SE) |
|---|---|---|---|---|---|
| Age at First Birth | 0.1657 (0.0073) | 0.2135 (0.0143) | $1.28 \times 10^{-5}$ | 1.035 (0.010) | 1.015 (0.008) |
| BMI | 0.2281 (0.0065) | 0.2381 (0.0091) | 0.14 | 1.127 (0.015) | 1.033 (0.011) |
| Breast cancer | 0.0259 (0.0034) | 0.0512 (0.0059) | $2.37 \times 10^{-8}$ | 1.021 (0.008) | 0.985 (0.007) |
| Drinks per Week | 0.0599 (0.0030) | 0.0739 (0.0054) | $7.44 \times 10^{-4}$ | 1.005 (0.008) | 0.985 (0.006) |
| Height | 0.4235 (0.0189) | 0.4464 (0.0206) | 0.059 | 1.479 (0.035) | 1.169 (0.020) |
| Physical activity | 0.0281 (0.0019) | 0.0311 (0.0044) | 0.408 | 0.996 (0.007) | 0.993 (0.007) |
| Self-rated health | 0.0972 (0.0029) | 0.1250 (0.0052) | $9.35 \times 10^{-13}$ | 1.052 (0.010) | 1.009 (0.008) |
| Severe Obesity | 0.0416 (0.0022) | 0.0584 (0.0045) | $1.83 \times 10^{-6}$ | 1.017 (0.008) | 0.995 (0.008) |
| Type 1 Diabetes | 0.0054 (0.0014) | 0.0432 (0.0035) | $1.63 \times 10^{-41}$ | 1.019 (0.007) | 0.940 (0.006) |
| Years of Education | 0.1482 (0.0052) | 0.1775 (0.0073) | $2.07 \times 10^{-9}$ | 1.164 (0.016) | 1.053 (0.011) |

SNP-based heritabilities for GWAS (column 1) and WGWAS (column 2) were estimated using LD-score regression (see Methods). The third column shows the *p*-value for the null hypothesis that the GWAS and WGWAS heritabilities are the same, based on a two-sided *Z*-test (see Methods). The fourth and fifth columns show the intercept of the LD-score regression in GWAS and WGWAS, respectively. An intercept >1 can be attributed to bias arising from population stratification[22].

### Volunteer bias affects gene tissue expression results

Gene tissue expression analyses exhibit different results for various traits in WGWAS, compared to GWAS. Hence, ignoring volunteer bias when estimating GWAS may result in a biased understanding of the biological pathways through which genes operate on a phenotype. Here, we highlight the results for breast cancer (Fig. 3). For this

phenotype, unweighted GWAS results show no evidence of genes expressed in any particular area of the body, to be significantly more associated with the likelihood of breast cancer. However, when estimating the same associations through WGWAS, we find that genes expressed in the fallopian tube, uterus, and ovary are more likely to exhibit associations with breast cancer. Thus, correcting GWAS for

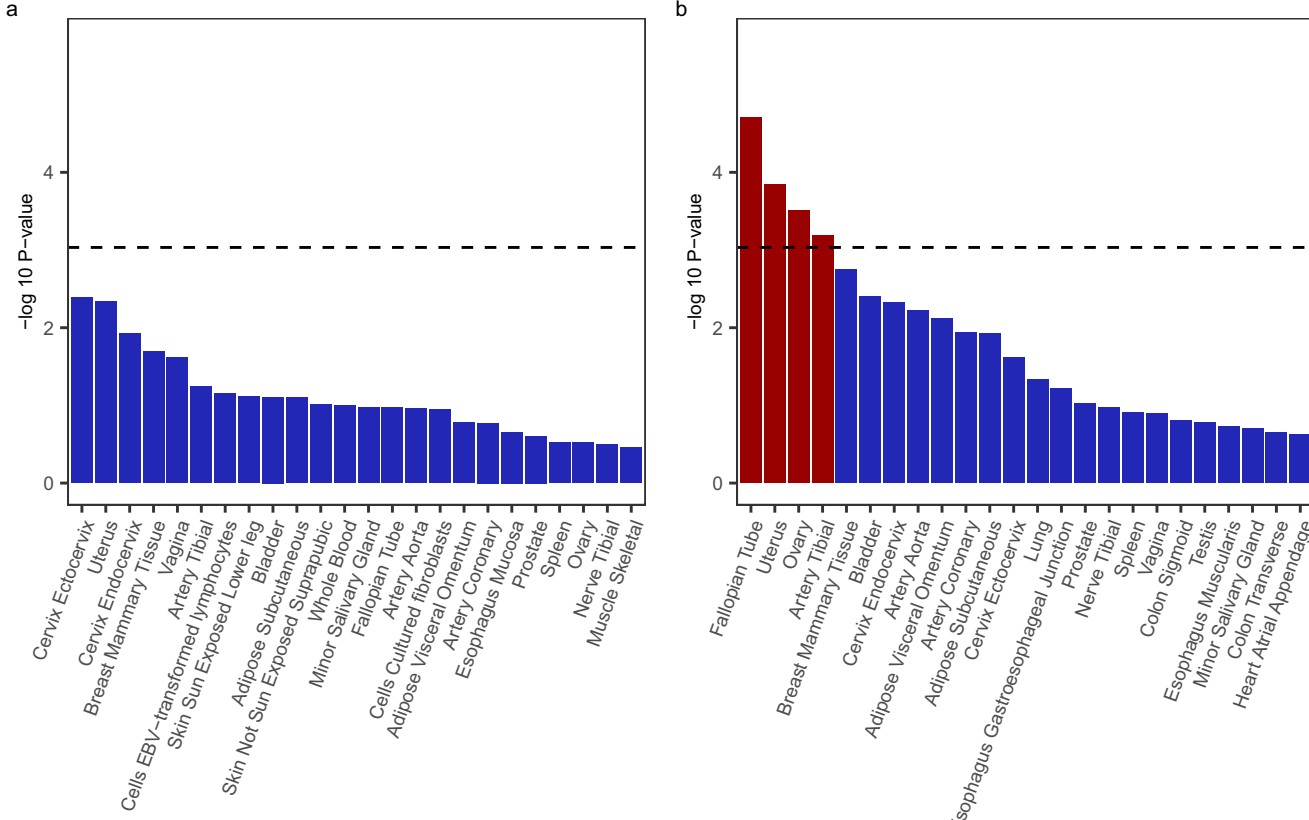

**Fig. 3 | Gene-set analysis for Breast Cancer, estimated using MAGMA, for GWAS and WGWAS, across 54 gene sets.** Panel (**a**) displays the results as estimated by GWAS. Panel (**b**) displays the results as estimated by WGWAS. Only the 15 gene sets with the lowest *p*-value are included in the plots. The dotted horizontal line denotes the 5% Bonferroni-corrected significance level (correction for 54 hypotheses). *P*-values were calculated using a one-sided test, where the null hypothesis states that the gene set does not exhibit above-average expression for the trait.

volunteer bias may improve understanding of the pathways through which the genome influences a phenotype of interest.

In supplementary material (Supplementary Figs. S6–S13), we show gene tissue expression analyses for the 9 other phenotypes. We find several phenotypes for which areas of the body are significantly more expressed in GWAS, but not in WGWAS, namely for AFB, BMI, self-reported health, and physical activity, suggesting that such GWAS findings might be spurious and driven by volunteer bias.

## Discussion

Our analyses highlight the drawbacks of non-random, volunteer-based sampling for GWASs and subsequent downstream genetic analyses. Contrasting WGWAS with GWAS results for ten phenotypes, we demonstrated that in GWASs, volunteer bias results in (i) attenuated SNP-effect sizes, (ii) missing heritability, (iii) biased gene-tissue expression findings, and (iv) missing potentially genome-wide significant loci. Our results suggest the effect of volunteer bias on GWAS is phenotype-specific. Phenotypes for which weighting altered results substantially were disease-related (e.g., T1D, breast cancer), related to socioeconomic status (e.g., education), or related to health behaviour (e.g., drinks per week). By contrast, for anthropomorphic phenotypes (height and BMI), weighting made a relatively minor difference. Although weighting still altered various GWAS-derived results for height and BMI, researchers may wish to opt for GWAS (rather than WGWAS) for such phenotypes, because of a bias-variance tradeoff, which increases the standard errors of WGWAS vis-à-vis GWAS.

Our results provide insights into the effects of volunteer bias on GWAS, but drawbacks remain. The IP weights we use to correct for volunteer bias may suffer from omitted variable bias since the model used to create them is limited to variables that the UKB and UK Census have in common. These variables mostly capture socioeconomic status, demographics, and self-reported health. It is possible other important variables relating to UKB volunteering are missing, e.g., personality characteristics. One indication that the weights do not capture the full extent of volunteer bias is the fact that sex remains significantly heritable on the autosome, even after conducting WGWAS (although the estimated heritability decreased after weighting from 1.13% [$p < 1 \times 10^{-8}$] to 0.95% [$p = 0.0015$]; see Supplementary Note S6). Nonetheless, these weights have been shown to substantially reduce bias, capturing an average of 87% of volunteer bias in phenotype-phenotype associations[3]. Therefore, we consider our analyses as indicative of pervasive volunteer biases in GWASs, with the true extent of bias probably larger than demonstrated here. In the presence of omitted variable bias in the weights we developed, differences between unweighted GWAS results and true underlying sampling population estimates could be even more pronounced than the substantial effects we already demonstrated here.

Our work builds on other studies that have considered weighting. In particular, a recent study constructed weights based on the HSE[18]. Some of our results are in contrast to these previous findings. For example, we find that heritability for various traits increases after weighting, whereas these authors find no statistically significant change in heritability or that it decreases. We attribute these differences to (1) a larger sample size (we can weight the full UKB), (2) more precisely estimated weights based on 687,489 UK Census respondents[3] versus their 22,646 HSE respondents, (3) our use of important predictors such as the region in weight estimation, and (4) the UK Census observations being more relevant: we were able to

make these more representative of the target population of the UKB (those living closely to one of 22 assessment centres), which, due to the non-random location of UKB assessment centres, is not the same as the population of England as sampled by HSE. As a result, the weights used here, estimated using large, fine-grained UK Census data on many variables that relate to participation, are the best possible weights available to capture volunteer bias in the UKB. The difference in performance between the IP weights we used compared to the HSE-based weights is reflected in the fact that our weights have substantial SNP-based heritability, 4.8% (s.e. 0.8%), compared to 0.9% (s.e. 0.5%) for the HSE-based weights[18]. We think the advantages of our weights are distinctive and would encourage their use in future work.

The focus here was on the UKB. Many other GWAS cohorts are volunteer-based and may similarly suffer from volunteer bias. Our results suggest that volunteer biases need to be taken seriously and can be corrected. GWAS consortia are advised to ensure that weights are available for all volunteer-based cohorts included in their GWAS. Such IP weights can be estimated by comparing the genotyped data set to a source of representative data (e.g., Census data or administrative data), provided that both data sets have a sufficient number of (close to) identically measured variables in common. Further, in the design of a new data set, it is essential that as many variables as possible are collected that are shared with a source of representative data to ensure that IP weights can be precisely estimated. Our results suggest that IP weighting is sufficient to capture a substantial degree of volunteer bias in genetic association results. WGWAS increases standard errors but is also likely to increase effect sizes, such that power need not be reduced. Further, WGWAS reduces the effective sample size of a cohort, which should be taken into account when meta-analysing multiple cohorts.

A potential limitation is that none of our results (GWAS and WGWAS) correct for potential cryptic relatedness. Mixed models are often used to deal with cryptic relatedness in the UKB, as a large number of UKB participants exhibit third-degree relatedness or more[26]. We did not remove third-degree relatives in UKB, as this would greatly reduce our sample size. Current software packages used to estimate mixed models on large-scale genetic data, such as BOLT-LMM or fastGWAS[19,27] are not designed to include sample weights. As such, estimating WGWAS using a linear mixed model is currently infeasible. Therefore, we estimated our models (GWAS and WGWAS) removing only second-degree relatives, without taking any residual cryptic relatedness into account. To test whether differences between GWAS and WGWAS results for ten phenotypes were not driven by cryptic relatedness, we estimated GWAS on the IP weights using a linear mixed model (correcting for cryptic relatedness) and compared its results to our more simple linear model (not correcting for cryptic relatedness). GWAS results on IP weights estimated using a linear mixed model did not differ from those estimated using the simple linear model, suggesting that cryptic relatedness in the IP weights is negligible. Because we systematically applied the same methodology to our GWAS and WGWAS results, cryptic relatedness does not appear to affect the comparisons made between GWAS and WGWAS presented in this paper, as cryptic relatedness likely influences the phenotype equally in both analyses. We encourage developers of GWAS software to include a sample weights option in software packages in the future.

In sum, our results suggest revisiting the current state of GWAS analyses may be warranted and has potential payoffs. Such revisiting could be based on carefully constructed population-representative data sets. This could be achieved through careful development of IP weights by ensuring relevant variables are collected that are shared with population-representative datasets, such as a population Census. Alternatively, a greater focus on population-representative sampling is advised.

## Methods

### Data

**UK Biobank (UKB).** The UKB is a cohort of 503,317 individuals collected between 2006 and 2010 at 22 assessment centres spread out across Great Britain. Potential participants were identified through the registry of the National Health Service, which covers virtually the whole UK population. Individuals living in proximity to an assessment centre and aged 40 to 69 at the start of the assessment period (which varies per assessment centre) received an invitation to participate by post. This UKB-eligible population consists of 9,238,453 individuals who received an invite, such that the overall acceptance rate was 5.45%.

Supplementary Fig. S1 summarises our sample selection criteria for the UKB. We drop individuals that were not included in the genetic subsample and restrict the UKB to individuals who identified as white British and were of genetic European ancestry, as most published work with the UKB genetic data uses this sample restriction (e.g., refs. [23,24,28]). We also dropped respondents who did not meet the standard requirements regarding genetic data quality control (see next subsection). Last, we dropped 6,292 respondents (1.6%) for whom IP weights could not be estimated, typically because of missing variables (see ref. [3] for details)

**Genetic data in the UKB.** Genetic data collection on UKB participants has been extensively described elsewhere[26]. We restrict our sample to those of white British ancestry, as defined by a PC analysis conducted by Bycroft et al.[26] As is the standard in GWAS analyses, we only keep UKB participants that were sufficiently densely genotyped: we drop individuals that have missing values at more than 2% of all SNPs measured in UKB (6118 participants in total). We also drop those with outlying heterozygosity values (mean +/− 3 std. deviations of the heterozygosity distribution observed in the data; 2279 participants in total). Furthermore, we drop individuals for whom their reported sex does not match with their sex as inferred from their measured genome (296 in total), as such mismatch may point towards sample contamination or sample mix-up. We focus on a genotyped sample that is approximately independent by keeping only one individual from each group of first-degree relatives. The individual that is kept is the one with the least missingness in their genetic data. Combined, we dropped 18,736 respondents from the sample.

We conduct our analyses on autosomal SNPs which are in HWE ($p > 1 \times 10^{-6}$), with MAF > 0.01, and which are missing in less than 2% of all included respondents, as recommended in ref. [29]). For reasons of computational feasibility, we restrict our analyses to 1,025,058 autosomal SNPs identified in HapMap3 that were available in this UKB-imputed genotyped data set[30]. Although regular GWAS typically examine a more extensive collection of SNPs, HapMap3 offers comprehensive coverage across the entire genome. In addition, numerous post-GWAS analytical tools, such as LD-score regression, only focus on the HapMap3 subset. Therefore, limiting our analysis to HapMap3 SNPs adequately illustrates how selection biases affect GWAS outcomes.

### IP weight estimation

The IP weights that we used were estimated in a previous study[3]. This study used the 5% random subsample of the 2011 UK Census to construct a subsample of the UKB's target population. The UKB's target population was defined as those who fulfilled the criteria to receive an invitation to participate in the UKB. Specifically, those who lived within an assessment centre's sampling range (typically less than 40 km around an assessment centre) and who were of the right age (40−69 years during the period of assessment [2006-2010]). This resulted in a target sample of 687,491 individuals.

Using this sample as the UKB's target population and the UKB sample itself, estimates of the probability to participate in the UKB were estimated using a probit model with LASSO variable selection, to

prevent overfitting of the data. Variables included in weight estimation were the year of birth, sex, educational attainment, employment status, region of residence, tenure of dwelling, number of cars in the household, self-reported health, and a single-household indicator. We considered these variables as categorical predictors in the model and included all possible two-way interactions. From the estimated participation probabilities, weights inversely proportional to the probability of participation were constructed. Last, to avoid extreme values of weights due to estimation uncertainty, the IP weights were winsorized: values lower than the 1st percentile were set equal to the value of the 1st percentile, and values higher than the 99th percentile were set equal to the 99th percentile.

## GWAS on the IP Weights

We estimate a GWAS using the IP weights as a phenotype by fitting a linear model in PLINK 1.9, restricting to our quality-controlled set of HapMap3 SNPs[31]. Note that, in our main analysis, we did not include any control variables, as any association between the genetic markers and volunteering propensity, whether this association is direct or indirect, could result in bias in typical GWAS. Hence, the goal here is to study associations between SNPs and the IP weights that are both direct (i.e., causal) and indirect (i.e., driven by population stratification, environmental confounding, assortative mating or genetic nurture). Independent hits of the GWAS on the IP weights were assessed through PLINK's clumping algorithm ($R^2 \geq 0.1$, LD-window of 250 kb). SNP-based heritability was estimated using LD-score regression[22] (see subsection 4.8 for additional detail). To assess the genetic overlap between the IP weights and various other traits, LD-score regression was used to estimate genetic correlations between the GWAS results on the IP weights and publicly available GWAS results for various phenotypes (see Supplementary Data 3), again estimated using LD-score regression.

To test the extent to which signal in this GWAS was driven by population stratification, we also report the heritability from a GWAS that controls for our regular set of control variables (see Regular GWAS and WGWAS), which crucially includes the first 20 principal components of the genetic data to control for population stratification[32]. This GWAS was estimated twice: once in PLINK 1.9, and once in BOLT-LMM, where the latter constructs a genomewide relatedness matrix to control for cryptic relatedness[19]. This genome-wide relatedness matrix was created using BOLT-LMM on the same set of SNPs used in creating the PGI repository (pruned SNPs with $r^2 < 0.3$, MAF > 0.01, and INFO > 0.6)[33]. Last, we estimated this heritability in a weighted GWAS (with these control variables), according to the estimation procedure described in subsections Regular GWAS and WGWAS and SNP-based heritabilities and genetic correlations.

## Regular GWAS and WGWAS

For each phenotype, we estimate GWAS associations for all HapMap 3 SNPs that were available in the UKB data. We fit the following model:

$$\tilde{y}_i = \alpha + \beta \text{SNP}_{ij} + \varepsilon_i, \tag{1}$$

where $\tilde{y}_i$ is the estimated residual of the phenotype from an auxiliary regression which fits $y_i$ on a set of variables that may confound the relationship between $SNP_j$ and $y$. These variables are genetic sex, the first 20 principal components of the genetic data, genotype measurement batch fixed effects, and a dummy for individual $i$'s birth year cohort (5-year bins) capturing the effects of aging on $y_i$. $\text{SNP}_{ij}$ is the individual $i$'s allele count at the $j$th SNP.

We estimate two GWASs for each phenotype: (1) a regular GWAS, which estimates SNP associations using the above model by OLS, and (2) an inverse probability-weighted GWAS (WGWAS), which estimates the above model using the IP weights that correct for volunteer bias as estimated in ref. 3, through weighted least squares. For WGWAS, $\tilde{y}$ was

residualized using the same IP weights in the auxiliary regression. We estimate heteroskedasticity-robust (White) standard errors for both GWAS and WGWAS[34]. Both GWAS and WGWAS were estimated in R. The resulting association estimates are denoted $\hat{\beta}^{GWAS}$ and $\hat{\beta}^{WGWAS}$ respectively.

## Comparing GWAS and WGWAS results for known top hits

Known top hits were selected from publicly available GWAS results that did not include the UKB as part of their discovery sample (See Supplementary Data 4), which were available for 6 out of 10 phenotypes. We selected top hits in this fashion and not using, e.g., our own UKB GWAS analyses to ensure that the selected top hits were not artificially overestimated due to the winner's curse[35]. To obtain top hits that were approximately independent, we clumped these results using PLINK ($R^2 \geq 0.1$, LD-window of 250 kb). Top hits were selected by only selecting SNPs with a cutoff $p < 10^{-5}$.

## Testing for significant differences in WGWAS and GWAS associations

We test the null hypothesis that the estimates of $\beta$ in Equation (1) as obtained through GWAS and WGWAS are the same, i.e., $H_0 : \hat{\beta}^{GWAS} = \hat{\beta}^{WGWAS}$, by constructing a Hausman test statistic: $H = \frac{(\hat{\beta}^{GWAS} - \hat{\beta}^{WGWAS})^2}{\mathbb{V}(\hat{\beta}^{GWAS} - \hat{\beta}^{WGWAS})}$, where $\mathbb{V}$ denotes the variance. In this expression, we use $\mathbb{V}(\hat{\beta}^{GWAS} - \hat{\beta}^{WGWAS}) = \mathbb{V}(\hat{\beta}^{GWAS}) - \mathbb{V}(\hat{\beta}^{WGWAS})$, given that $\hat{\beta}^{GWAS}$ is estimated efficiently under the null[36,37]. Estimates of $\mathbb{V}(\hat{\beta}^{GWAS})$ and $\mathbb{V}(\hat{\beta}^{WGWAS})$ are easily approximated by squaring the standard errors of $\hat{\beta}^{GWAS}$ and $\hat{\beta}^{WGWAS}$, respectively. This test statistic follows a chi-squared distribution with 1 degree of freedom. Hence, $P$-values for rejection of the null hypothesis (denoted $P_H$) are obtained by comparing H to this chi-squared distribution.

## Determining the effective sample sizes of GWAS and WGWAS

The effective sample size aids in understanding how much non-representativeness dilutes the power of GWAS results and are a crucial input into the LD-score regressions (see next subsection). We calculate the effective sample size for each SNP[21], given by

$$N_{eff} = \frac{\sigma_{y,k}^2}{SE_k^2 \cdot [2 \cdot MAF_k \cdot (1 - MAF_k)]}, \tag{2}$$

with $k \in GWAS, WGWAS$ referring to either the unweighted or IP weighted sample statistic, $\sigma_{y,k}^2$ the variance of the phenotype, MAF the minor allele frequency of the SNP, and $SE_k^2$ the standard error of the SNP as determined by unweighted or IP weighted GWAS, respectively. For each phenotype, the effective sample size as averaged over all SNPs, is reported.

## SNP-based heritabilities and genetic correlations

We use LD-score regression to estimate the genetic correlation and SNP-based heritabilities for GWAS and WGWAS[22,38]. GWAS and WGWAS summary statistics were prepared using the `munge_sumstats.py` function of the ldsc package[22]. Our estimates of $N_{eff}$ were used as the parameter for the sample size when preparing the summary statistics for both GWAS and WGWAS. Some research suggests that, for binary phenotypes, a transformation towards the liability scale is necessary to interpret SNP-based heritabilities properly[39]. This scale needs the population prevalence as an additional parameter. However, since our goal is not to make definitive statements about true SNP-based heritabilities, but rather to compare GWAS with WGWAS, we decide to report these heritabilities on the observed scale (i.e., without

correction for population prevalence). Thus, a comparison between GWAS and WGWAS results can be made based on the estimated associations of the SNPs, and not based on additional changes in estimates of population prevalence.

To evaluate whether our SNP-based heritabilities differed for GWAS and WGWAS, we construct the following Z-statistic:

$$Z = \frac{h^2_{GWAS} - h^2_{WGWAS}}{\sqrt{s.e.(h^2_{GWAS}) + s.e.(h^2_{WGWAS}) - 2cov(h^2_{GWAS}, h^2_{WGWAS})}}, \quad (3)$$

with $h^2_{GWAS}$ and $h^2_{WGWAS}$ the SNP-based heritabilities estimated through GWAS and WGWAS, respectively, $s.e.(h^2_{GWAS})$ and $s.e.(h^2_{WGWAS})$ their standard errors, and $cov(h^2_{GWAS}, h^2_{WGWAS})$ the covariance of these estimates, which is computed by

$$cov(h^2_{GWAS}, h^2_{WGWAS}) = cor(h^2_{GWAS}, h^2_{WGWAS}) \times s.e.(h^2_{GWAS}) \times s.e.(h^2_{WGWAS}),$$

estimating $cor(h^2_{GWAS}, h^2_{WGWAS})$ as the value of the intercept from the cross-trait LD-score regression on the weighted and unweighted GWAS results[21,38].

### Gene tissue expression analyses

Gene tissue expression analysis is a popular tool for understanding the biological pathways through which genes may operate on a phenotype. We assessed the relevance of volunteer bias for such bio-annotations by conducting gene-set analyses using our WGWAS and GWAS summary statistics in MAGMA (implemented through the FUMA pipeline)[40,41]. This pipeline evaluates the enrichment of genetic associations across 54 gene sets, categorised by tissue type, to determine if this is significantly higher than average.

### Reporting summary

Further information on research design is available in the Nature Portfolio Reporting Summary linked to this article.

## Data availability

UK Biobank data is accessible upon request and approval by the UK Biobank committee (https://www.ukbiobank.ac.uk/). The IP weights developed here are available in the returned results catalogue under appliciation number 55154. Summary statistics of all GWAS and WGWAS analyses mentioned in this paper are available at https://github.com/sjoerdvanalten/UKB_WGWAS/tree/main/GWAS_Final and https://github.com/sjoerdvanalten/UKB_WGWAS/tree/main/WGWAS_Final, respectively.

## Code availability

All code used for generating the results is available at https://github.com/sjoerdvanalten/UKB_WGWAS[42].

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

## Acknowledgements

Research reported in this publication was supported by the European Union's Horizon 2020 research and innovation programme under the Marie Skłodowska-Curie grant agreement (ESSGN 101073237), the National Institute on Aging of the National Institutes of Health (RF1055654, R56AG058726, R01AG078522, and R01AG079554), the Dutch National Science Foundation (016.VIDI.185.044), and the Jacobs Foundation. This research has been conducted using the UK Biobank Resource under Application Number 55154. We thank participants at the 2021 and 2022 BGA annual meetings, the 2021 and 2022 ASHG conferences, the 2021 and 2023 Integrating Genetics and Social Science Conference, and the 2022 European Social Science Genetics Network Conference for their feedback and comments. We thank Aysu Okbay for the valuable comments and sharing of a list of pruned UKB SNPs. We also thank Michel Nivard, Ronald de Vlaming, Hyeokmoon Kweon, Elliot Max Tucker-Drob, and Wei Zhao for their kind feedback and valuable comments.

## Author contributions

S.A. was responsible for all data analysis. ATM checked the coding and data analysis process. S.A. was responsible for the first draft of the manuscript. S.A., B.W.D., J.F., T.J.G and A.T.M. were jointly responsible for designing the study, draughting the final manuscript, and revising its contents.

## Competing interests

The authors declare no competing interests.
