## [Transparent Peer Review file · Nature Communications]

Correcting for volunteer bias in GWAS increases SNP effect sizes and heritability estimates

Corresponding Author: Mr Sjoerd van Alten

Version 0:

Reviewer comments:

Reviewer #2

(Remarks to the Author)

The paper has improved a lot and the authors have answered to most of my concerns.

I still have a single concern: the authors are still not accounting properly for population structure in their analyses. They have added 20 genetic PCs to their models which capture part of the structure but will not capture family relationships. The proper way would be to use either a mixed model approach or WGR as implemented in regenie.

The fact that the weights partly account for this is not enough as this should be accounted for completely. The removal of one of the two siblings when present is not enough to eliminate the problem completely.

This may also explain the SNPs which show up for T1D and Breast Cancer only in their GWAS and not in any other. These methods are standard in GWAS studies and should always be applied.

Finally, it seems that the GWAS results have not been deposited anywhere. Also, the code makes references to files which aren't provided. I appreciate that this is not always possible but to make this reproducible the comments should mention what the input files contain and look like.

(Remarks on code availability)

See comments to authors.

Version 1:

Reviewer comments:

Reviewer #2

(Remarks to the Author)

The authors have addressed all final comments and I have no more concerns.

(Remarks on code availability)

Dear referee,

Thank you very much for taking the time to review our paper “Correcting for volunteer bias in GWAS increases SNP effect sizes and heritability estimates” in detail. Based on the feedback you provided, we have revised specific parts of the paper. We believe the new version fully takes your concerns into account. Below, we provide a point-to-point response to your feedback, clearly indicating which changes to the paper were made in order to address your concerns. Novel and changed sentences to the paper are also highlighted in the pdf, at your convenience.

Kind regards,

Sjoerd van Alten and co-authors.

Point-to-point response

The paper has improved a lot and the authors have answered to most of my concerns.

I still have a single concern: the authors are still not accounting properly for population structure in their analyses. They have added 20 genetic PCs to their models which capture part of the structure but will not capture family relationships. The proper way would be to use either a mixed model approach or WGR as implemented in regenie. The fact that the weights partly account for this is not enough as this should be accounted for completely. The removal of one of the two siblings when present is not enough to eliminate the problem completely.

This may also explain the SNPs which show up for T1D and Breast Cancer only in their GWAS and not in any other.

These methods are standard in GWAS studies and should always be applied.

We thank the reviewer for this comment. In response, we have used BOLT-LMM (accounting for cryptic relatedness) to investigate whether cryptic relatedness is a spurious driver of associations between SNPs and the IP weights. Using BOLT-LMM to estimate a GWAS on the IP weights, we obtained near-identical results compared to using a simple linear model implement in PLINK 1.9 ($r_G = 1.0009$, $s.e. = 0.0018$). Hence, we conclude cryptic relatedness is likely of little concern when comparing results estimated using GWAS and WGWAS. We have added the following to section 2.1:

“To verify that any remaining heritability was not inflated due to residual population stratification or cryptic relatedness, we accounted for these by re-estimating this GWAS using a linear mixed model (BOLT-LMM),¹⁹ ($h^2=2.24\%$, $s.e.=0.16\%$). Accounting for residual population stratification and cryptic relatedness did not alter our results. The genetic correlation of the GWAS results estimated with the

linear model, using PLINK, and the linear mixed model, using BOLT-LMM, was indistinguishable from 1 ($r_G=1.0009$, $s.e.=0.0018$).”

Further details are added to the methodology sections, new sentences underlined (section 4.3):

“To test the extent to which signal in this GWAS was driven by population stratification, we also report the heritability from a GWAS that controls for our regular set of control variables (see subsection 4.4), which crucially includes the first 20 principal components of the genetic data to control for population stratification.⁹ This GWAS was estimated twice: once in PLINK 1.9, and once in BOLT-LMM, where the latter constructs a genomewide relatedness matrix to control for cryptic relatedness.¹⁰ This genomewide relatedness matrix was created using BOLT-LMM on the same set of SNPs used in creating the PGI repository (pruned SNPs with $r^2 < 0.3$, $MAF > 0.01$, and $INFO > 0.6$).¹¹”

The lack of evidence for cryptic relatedness in the IP Weights is a strong indicator that differences between results estimated in GWAS and WGwas are not inflated due to differences in cryptic relatedness between the two analyses. A more direct test to control for cryptic relatedness would be to estimate all GWAS and Weighted GWAS results (on all 10 phenotypes) using a linear mixed model rather than a (weighted) linear regression. Unfortunately, we are not aware of any mixed modelling software that currently allows for the inclusion of sample weights. Developing such an option is non-trivial given the computational complexity of such models. We now mention this as a limitation in the discussion, and encourage developers of statistical software packages to allow for inclusion of sample weights in the future:

“A potential limitation is that none of our results (GWAS and WGwas) correct for potential cryptic relatedness. Mixed models are often used to deal with cryptic relatedness in the UKB, as a large number of UKB participants exhibit third-degree relatedness or more²⁶. We did not remove third-degree relatives in UKB, as this would greatly reduce our sample size. Current software packages used to estimate mixed models on large-scale genetic data, such as BOLT-LMM or fastGWAS^{19,27} are not designed to include sample weights. As such, estimating WGwas using a linear mixed model is currently infeasible. Therefore, we estimated our models (GWAS and WGwas) removing only second-degree relatives, without taking any residual cryptic relatedness into account. To test whether differences between GWAS and WGwas results for ten phenotypes were not driven by cryptic relatedness, we estimated GWAS on the IP weights using a linear mixed model (correcting for cryptic relatedness), and compared its results to our more simple linear model (not correcting for cryptic relatedness). GWAS results on IP weights estimated using a linear mixed model did not differ from those estimated using the simple linear model, suggesting that cryptic relatedness in the IP weights is negligible. Because we systematically applied the same methodology to our GWAS and WGwas results, cryptic relatedness does not appear to affect the comparisons made between GWAS and WGwas presented in this paper, as cryptic relatedness likely

influences the phenotype equally in both analyses. We encourage developers of GWAS software to include a sample weights option in software packages in the future.”

Finally, it seems that the GWAS results have not been deposited anywhere. Also, the code makes references to files which aren't provided. I appreciate that this is not always possible but to make this reproducible the comments should mention what the input files contain and look like.

We thank the reviewer for this comment and apologize for the previous lack of clarity in the code. We have updated the README of the Github repository to provide more detail on the files needed to reproduce our results. Unfortunately, we cannot provide these files directly, given the restricted nature of UKB data. However, we can share the GWAS and WGWAS summary statistics freely. These are now also available in the Github repository. Also, UKB is now providing the inverse probability weights as well. The data availability statement in our paper has been adjusted accordingly:

“UK Biobank data is accessible upon request and approval by the UK Biobank committee (<https://www.ukbiobank.ac.uk/>). The IP weights developed here are available in the returned results catalogue under application number 55154. Summary statistics of all GWAS and WGWAS analyses mentioned in this paper are available

at https://github.com/sjoerdvanalten/UKB_WGWAS/tree/main/GWAS_Final and

https://github.com/sjoerdvanalten/UKB_WGWAS/tree/main/WGWAS_Final, respectively.”